# Accuracy of Measuring Knee Flexion after TKA through Wearable IMU Sensors

**DOI:** 10.3390/jfmk6030060

**Published:** 2021-07-05

**Authors:** Ricardo Antunes, Paul Jacob, Andrew Meyer, Michael A. Conditt, Martin W. Roche, Matthias A. Verstraete

**Affiliations:** 1EnMovi Ltd., Unit 1I, Inovo Building, 121 George St., Glasgow G1 1RD, UK; ricardo.antunes@enmovi.com; 2Oklahoma Joint Reconstruction Institute, 9800 Broadway Ext., Suite 201, Oklahoma City, OK 73114, USA; drjacob@drpauljacob.com; 3OrthoSensor Inc., Suite A-310, 1855 Griffin Rd., Dania Beach, FL 33004, USA; andrew.meyer@orthosensor.com (A.M.); michael.conditt@orthosensor.com (M.A.C.); 4Holy Cross Hospital, 4725 N Federal Highway, Fort Lauderdale, FL 33308, USA; martin@mroche.com

**Keywords:** knee arthroplasty, wearable sensors, IMU sensors, flexion measurements, accuracy

## Abstract

Wearable sensors have the potential to facilitate remote monitoring for patients recovering from knee replacement surgery. Using IMU sensors attached to the patients’ leg, knee flexion can be monitored while the patients are recovering in their home environment. Ideally, these flexion angle measurements will have an accuracy and repeatability at least on par with current clinical standards. To validate the clinical accuracy of a two-sensor IMU system, knee flexion angles were measured in eight subjects post-TKA and compared with other in-clinic angle measurement techniques. These sensors are aligned to the patients’ anatomy by taking a pose resting their operated leg on a box; an initial goniometer measurement defines the patients’ knee flexion while taking that pose. The repeatability and accuracy of the system was subsequently evaluated by comparing knee flexion angles against goniometer readings and markerless optical motion capture data. The alignment pose was repeatable with a mean absolute error of 1.6 degrees. The sensor accuracy through the range of motion had a mean absolute error of 2.6 degrees. In conclusion, the presented sensor system facilitates a repeatable and accurate measurement of the knee flexion, holding the potential for effective remote monitoring of patients recovering from knee replacement surgery.

## 1. Introduction

Patients recovering from knee replacement surgery are often monitored by various stakeholders (physiotherapists, surgeons, nurse practitioners, etc.) who each get a different snapshot of the state of recovery at different time intervals. However, the overarching view of the temporal progression of recovery is often missing, which limits the granularity of the feedback to both the surgeon and the physiotherapist. More frequent feedback may be necessary to understand the true impact of their clinical/surgical decisions and improve the currently observed, suboptimal patient-reported outcomes in knee arthroplasty surgery [1,2].

Knee range of motion (ROM) is an important parameter in evaluating outcomes and recovery after total knee arthroplasty (TKA). In standard practice, knee ROM is assessed by clinical practitioners, often with the patient in a supine position. The extremes of terminal extension and maximum flexion are of primary clinical relevance, both active (i.e., driven by the patient’s muscular ability [3]) and passive (i.e., under external force by a physiotherapist [4]). The accuracy of the knee flexion measurements directly affects these assessments. A range of angle measurement methods have been reported, including the use of radiographic goniometry, marker-based optical motion capture, visual estimation, hand goniometry (short and long arm goniometer), videography [5] and digital goniometers [3]. Radiography and marker-based optical motion capture are considered the most accurate kinematic measurement tools, though both are seldom used in clinical practice. Radiation exposure for the purpose of goniometry is often considered excessive and the instrumentation and data processing associated with marker motion capture results in significant overhead costs. The alternative methods have a high interrater and intrarater reliability but vary in accuracy ranging from a 14° minimum significant difference for short arm goniometry to 6° when using a digital goniometer [3]. Regardless of which of the aforementioned methods is used in clinical practice, the measurements remain largely constrained to clinical visits at preset time intervals and do not facilitate continuous, remote patient monitoring.

Recent advances in wearable sensors provide an opportunity for (continuous) remote monitoring of patients’ knee flexion by the care team rather than discontinuous, momentary assessments by individual stakeholders [6]. Using two inertial measurement units (IMUs) placed above and below the knee, these sensors can measure knee flexion angles both at home and in a clinic. These angles can be subsequently uploaded to the cloud via a smartphone for clinicians to review. The use of these sensor systems holds the potential to measure knee metrics during activities of daily living as well as to track and guide patients’ home exercise physiotherapy program during recovery. However, accurate measurement of the knee flexion angle by these skin-mounted sensors is essential to achieve the above goals [7].

Accurate measurement of knee joint angles using IMUs requires proper sensor-to-leg registration, where the individual sensor axes are aligned with the underlying mechanical axes that define the knee flexion angle (i.e., the axis from the hip to the knee and the knee to the ankle) [8]. Inaccurate registration of the sensors to the leg’s mechanical axes will introduce bias into the reported knee joint angle. This link potentially needs to be re-established every time sensors are re-attached to the patient’s leg, for example following an overnight recharging cycle. Furthermore, if sensors are intended for remote monitoring, patients should be able to perform sensor-to-leg registration without a clinician’s supervision at any point during their recovery. Several sensor-to-leg registration processes are in use today, though these methods impose challenges when applied to (post-)TKA patients. In particular, registration methods that require movement of the leg have reduced accuracy when applied to obese patients due to skin motion relative to the bone [3,9]. Registration methods that involve the subject assuming a straight leg pose [3,10] may not be effective in subjects post-TKA, whose knees cannot necessarily achieve full extension [11]. Finally, registration methods that utilize measurements of knee angles by some external tool (e.g., a goniometer or smartphone) may be difficult for patients to use at home or without clinician assistance or supervision. It is therefore concluded that new sensor-to-leg registration methods shall be developed and tested to improve the accuracy of IMU-based knee angle measuring sensors in a post-TKA population.

In this pilot study, the accuracy and repeatability of knee angle measurements made by the MotionSense system is evaluated (EnMovi, Glasgow, UK). This system measures knee angles using two IMU sensors placed above and below the knee joint on the lateral aspect of the thigh and shank respectively. The MotionSense sensors are registered to the leg using a novel technique whereby the knee angle reported by the sensors is set to a known value as measured by a goniometer while the subject assumes a consistent static pose. The following aspects of the sensor system are assessed in this paper:How consistent and repeatable is the sensor-to-leg registration process? This question is applied to the two components of the leg registration process:
How consistent are the goniometer measurements made by the clinician?How consistent is the static pose taken by the subject during the sensor-to-leg registration process?How accurate are the sensor angles through a wide range of motion? This will be tested by comparing sensor angle measurements with goniometer angles, as well as angles from a markerless optical motion capture system.

Focusing on TKA recovery applications and taking into account the abovementioned variability of the current standard of care, the desired sensor accuracy threshold is 5 degrees.

## 2. Materials and Methods

### 2.1. MotionSense Platform

The MotionSense wearable sensors consist of two sensor nodes that communicate via Bluetooth with a mobile app installed on the patient’s phone. Each sensor includes a six degree of freedom IMU sensor sampled at 50 Hz and subsequently processed through Madgwick filtering to calculate the pitch and roll angle for each sensor node. This results in a static accuracy measured on a mechanical hinge of 0.1 degree (Appendix A). These sensors are attached above and below the knee joint on the lateral aspect of the thigh and shank, respectively, using a dual patch system. The first, weekly patch remains in contact with the skin for a longer duration. The second patch adheres the sensors to the weekly patch and is removed daily to allow overnight sensor charging. After re-applying the sensors to the leg each morning, the sensor orientation relative to the patient anatomy is unknown (Figure 1). The true knee flexion angle, α_Knee_, is defined as the angle between the mechanical axes of tibia and femur, m_T_ with respect to m_F_ (as shown in Figure 1). The relative orientation of these mechanical axes cannot be measured directly but is approximated by the angle read by the sensors attached to the respective segments (α_Femur_ with respect to α_Tibia_). This approximation is reflected by the sensor misalignment angles, δ_F_ and δ_T_, representing the difference between the mechanical axes of the underlying anatomy and the sensor axes. This leaves the following relationship between the true knee flexion angle and the sensor readings:α_Knee_ = (α_Femur_−δ_F_)−(α_Tibia_−δ_T_)(1)
or
α_Knee_ = α_Femur_−α_Tibia_−δ_Knee_(2)

Since the design goal of the device is to accurately evaluate the knee angle, there is no need to identify δ_F_ and δ_T_ individually; instead, the combined alignment error δ_Knee_ is determined. In the remainder of this paper, the latter is referred to as the offset angle. In practice, this offset is estimated by the device while the patient assumes a pose, resting their operated leg on a rigid box with a fixed height of 10 cm. While striking this pose, the sensors calculate the offset between their relative orientation (α_Femur_−α_Tibia_) and the knee flexion angle measured by the physiotherapist during the initial onboarding visit (α_PT_). Subsequently, this offset angle (δ_Knee_) is stored locally on the sensors to allow for an accurate evaluation of the knee flexion angle throughout the day (Figure 2). The sensors automatically detect when they are either intentionally or accidentally removed from the leg, after which the above procedure must be repeated by the patient to obtain an updated offset angle.

### 2.2. Clinical Design

During an IRB approved study, a total of 8 participants were enrolled in a longitudinal study where patients used the MotionSense system and accompanying mobile app for 10 consecutive days. All participants had recently undergone knee replacement surgery using the Triathlon CR system implanted with a robotic assisted arm (Stryker, Kalamazoo, MI, USA). The patients had an average time since surgery of 27 days (see Table 1) and were demographically representative for the population undergoing TKA surgery (BMI up to 35, mean age of 63) [12]. Only patients that had undergone unilateral TKA within the past 3 months and owned a smartphone were eligible for enrolment in the study. In addition, patients with major complications (e.g., infections, instability) were not eligible for enrollment nor were major complications encountered for the participants in the study. An overview of the patients enrolled in the study is shown in Figure 3.

The patients participated in two in-clinic sessions at least 10 days apart during which the patients were asked to repeatedly rest their operated leg on the box. In between repeats, the sensors were not removed from the leg, although the patients were asked to walk around the facility, and the offset angles from repeated poses were compared. In the remainder of this paper, the first 5 repeats per participant have been considered for analysis. In addition, a sub-group of 5 patients performed two physiotherapy exercises: during the initial visit, patients performed 10 repeats of a standing knee bend, while during the return visit, they performed 10 standing knee bends and 10 long arc quads (Figure 4).

### 2.3. Video Motion Capture Measurements

To evaluate the accuracy and precision of the IMU-derived knee flexion measurements throughout the range of motion, the wearable sensor measurements were compared to angles from video motion capture during a series of exercises. A sagittal plane video (1920 × 1080 pixels, 60 fps) of the sensor-instrumented leg was recorded using a Nikon Z50 camera with a 16–50 mm lens mounted on a tripod. Subsequently, the exercise videos were processed to automatically detect the location of the patient’s hip, knee and ankle joints using the OpenPose library (version 1.6, Carnegie Mellon University, Pittsburgh, United States of America). OpenPose is a machine learning pose estimation library developed to detect human body key points in videos [13] (see Figure 5 for examples). OpenPose has been shown to have an almost complete agreement with joint angles obtained with a three-dimensional motion analysis system [14]. In addition, an internal validation was performed against traditional surface-marker-based motion capture analyses, showing excellent agreement between the OpenPose knee flexion angles and those measured using a surface-marker-based system (see Appendix B). This internal validation additionally indicated that OpenPose results can be sensitive to camera angle, body shape and the color and looseness of a subject’s clothing. As a result, the OpenPose results for all subjects were visually inspected for OpenPose sensitivity artefacts. Subjects were removed from the following analyses if it was apparent that OpenPose did not correctly identify the joint centers of interest.

Knee angles were calculated for each video frame as the internal angle between the hip-to-knee and ankle-to-knee vectors using custom Python 3.8 scripts. The video- and sensor-reported knee angles associated with the same exercise were subsequently resampled to a common sampling period using spline interpolation, corresponding to the video frame rate (i.e., 60 fps). In the next step, the resulting timeseries were synchronized by applying the sample offset that maximizes the covariance between video and sensor measurements. Any potential offset for the OpenPose data was then corrected for by matching the goniometer measurement while standing on the box, analogous to the process described earlier for the wearable sensors. The remaining bias between both timeseries can thus be seen as the dynamic accuracy through the range of motion of the wearable sensors relative to our validated video motion capture method. This bias is calculated from the mean deviation between time-aligned sensor and video knee angle measurements (sensor–video) within five-degree intervals. The bias between sensor and video knee angles was also assessed by fitting a linear regression model with aligned sensor measurements as a dependent variable and video angle measurements as the independent variable. To allow for correction of mean bias throughout the range of motion, the observed variation of bias was also modeled as a function of the sensor knee flexion angles. Because of the observed bias variation in different sections of the range of motion, a continuous piecewise linear function model was fitted. All data were processed using Python 3.8 including the pandas (v1.1.0) [15,16] framework for data handling and relying on scikit-learn (v0.23.1) [17] machine learning library for linear regression and the pwlf library (v2.0.4) [18] for continuous piecewise linear function model.

## 3. Results

### 3.1. Repeatability of Goniometer Measurements on Box

Each patient’s knee flexion angle while standing on the box was measured both during the onboarding and final visit in the clinic. During these measurements, the physiotherapist was blinded to the readings from the previous visit. The obtained knee flexion angles across all patients while taking repeats ranged between 50 and 74 degrees (Table 2). The mean absolute difference between the flexion angle measured during both visits was 2.6 degrees, with a standard deviation of 2.5 degrees.

### 3.2. Repeatability of Patient Pose on Box

Over both sessions, the patients performed five repeats resting their operated leg on the box (n). As the sensors remained on the patients’ legs during repeats, the resulting offset angles can be compared across repeats within a session to evaluate the variability in the patient assuming the pose. This variability is quantitatively assessed by removing the mean offset angle from the readings during a session. Overall, the mean absolute variability is 1.6 degrees with a standard deviation of 1.5 degrees (Table 3; Figure 6).

### 3.3. Linearity of Sensor Reading through Range of Motion

A total of 22 exercise sessions were logged for five patients that were matched with OpenPose video measurement timeseries. Patients 1, 2 and 8 were removed from this portion of the analyses since OpenPose failed to accurately identify their joint centers for a significant portion of the video. These exercises included 16 standing knee bends and four long arc quads (Table 4). The covariance criterion indicated good alignment between sensor and video knee flexion angle timeseries for all of the 22 logged sessions (e.g., Figure 5). The fitted linear models were statistically significant (*p* < 0.001) and with very good model fit (R^2^ > 0.99, example shown in Figure 7) for all individual exercise sessions as well as overall. The fitted slope values were all larger than, and very close to one (1.00–1.13 range), indicating that there is a good linear relationship between the sensor and video measurements on average. The 5-degree binned bias variation shown in Figure 8a indicates that measured sensor knee flexion angles have low bias between 30 and 70 degrees flexion, while slightly underestimating knee flexion angles above this range and slightly overestimating below 30 degree flexion. The 95-percentile range across the 5-degree bins ranged from 5.21 degrees for the 0 to 5-degree flexion bin to 15.28 degrees for the 105 to 110-degree flexion bin (see also Table 5).

### 3.4. Correction of Bias Non-Linearity

To allow for correction of mean bias at the extremes of the range of movement, the observed variation of bias was modeled as a function of the sensor knee flexion angles. Because of the observed bias variation in different sections of the range of motion, a continuous piecewise linear function model was fitted. The central point of the five-degree bins was used as the dependent variable (i.e., 0.5, 1.5, etc.) and the corresponding mean bias for each exercise session as the independent variable. This model was fitted with a fixed target number of three segments (i.e., 2 breakpoints) but without explicit breakpoint locations. This was chosen to preserve the low bias in the middle range, while compensating for the underestimation and overestimation at the extremes of the range of motion. The piecewise model fitted breakpoints at 15 and 47 degrees of flexion with an R^2^ = 0.36 (Table 6). Correcting the sensor angles with this model’s prediction (i.e., observed sensor–predicted bias) resulted in a reduction of the bias at the extremes of the ROM (Figure 8b and Table 5), as well as a reduction of the overall mean bias from −0.62 degree (SD 3.75) to −0.18 degree (SD 3.17).

## 4. Discussion

In this study, various aspects contributing to the accuracy and repeatability of an IMU-based wearable system in measuring knee flexion have been evaluated. Based on the data collected from a total of eight patients that recently underwent TKA surgery, the accuracy of the wearable system for measuring knee flexion was found to be below the desired five-degree threshold.

Because of limitations in currently available sensor-to-leg registration methods, we used a registration procedure where the reported sensor angles are corrected to a previously measured value while patients rest their operated leg on a box with known height. In this study, three aspects of the system accuracy linked to this registration method were assessed. First, the consistency of goniometer measurements underlying the sensor-to-leg registration method was evaluated. We found a mean absolute difference of 2.6 degrees in the angle measurements made by two clinicians several weeks apart while the patient takes a standardized static pose. Second, we evaluated the patients’ ability to repeatably take this static pose. We found that the knee angle while taking this pose had a mean absolute variability of 1.6 degrees. Third, the sensor accuracy through the range of motion was established by comparing the sensor readings to a video-analysis based method, after removing the overall bias. This comparison resulted in a mean absolute error through the range of motion of 2.6 degrees, with higher values observed in deep flexion and towards terminal extension. It was shown that this error can be corrected using a piecewise linear correction method to yield a mean absolute error through the range of motion of 2.3 degrees.

Evaluating a patient’s range of motion and the associated absolute knee flexion angles is critical for monitoring patients’ recovery progress as it has the potential to impact gait and function [19]. Current clinical practice provides limited opportunity to remotely monitor a patient’s range of motion; instead, the flexion range is primarily assessed during in-clinic visits with their physiotherapist or surgeon. The presented sensor system provides the opportunity to remotely monitor a patient’s range of motion on a daily basis while patients recover at home and carry out normal daily activities. Although beyond the scope of this paper, it is assumed that the associated clinical benefits are twofold. First, this sensor facilitates direct feedback to the patient during their home exercises. This can drive patient compliance with their home exercise program and assure proper execution of the exercises [20]. Second, the higher granularity of range of motion measurements can facilitate early detection of patients with suboptimal recovery (e.g., stiffness). This can drive early, conservative treatment changes over surgical interventions. In order to deliver on this promise, the sensor system needs to achieve the necessary accuracy as the data needs to be clinically relevant to the care team. The main aspects contributing to this system’s accuracy on a daily basis are the repeatability when the patients take the static pose during sensor-to-leg registration and the dynamic accuracy of the sensors through the range of motion. Our results indicated that this combined mean absolute error averages 4.2 degrees without applying corrections. This compares well to current clinical practice, where goniometer measurements or visual assessments remain the standard of care. For example, work by Edwards et al. [21] indicates that visual assessments can be off by over 5 degrees in 45% of cases, whereas goniometer measurements can be off by more than 5 degrees in as much as 22% of cases [3]. Based on the promising results from this pilot study, using the presented system to accurately monitor patients’ knee flexion angle in a remote setting will be explored in future work.

Previous studies focusing on the accuracy of IMU-based knee flexion measurement systems are often limited to the relative range of motion or based on assumptions around the patient’s ability to fully extend their leg or position sensors accurately. These studies often overlook the offset or systematic bias that could be introduced during the sensor-to-leg registration process studied in this paper and associated misalignment of the sensors relative to the patient’s anatomy [22,23]. This issue has important clinical limitations, particularly in a patient population recovering from knee replacement surgery or anterior cruciate reconstruction surgery [24] where a lack of full extension during recovery often leads to re-operation [25,26]. Our research is therefore unique in that it recognizes the dependency of knee flexion measurements on the datum position and considers the limitations of the targeted patient population. Through the presented study, our method of resting the operated leg on a rigid box has demonstrated to be feasible and repeatable. Previous research has shown that the knee flexion angles observed in our work while resting on the box (ranging between 58 and 74-degree flexion) are achievable from the early post-operative phase and throughout the recovery for TKA patients [27]. Therefore, the position on the box with fixed height can be seen as a reliable alternative for TKA patients compared to the neutral, straight standing pose often used in biomechanical research with healthy individuals [3].

The assessment of the dynamic accuracy through the range of motion revealed a small positive bias that was systematically observed at the lower end of the range of motion, while a small negative bias was observed at the higher end. This was driven by a slightly positive slope between our sensor readings and the video motion capture measurements. Should we have selected one extreme of the range of motion as the datum position, the maximum error would have been larger at the other end of the range of motion. Instead, the box calibration pose is based on a flexion angle near the middle of the range of motion, thus keeping the maximum errors limited at these extreme ends which are often of primary interest to monitor patient recovery [10]. Indeed, the mean error through the range of motion was small at −0.52 degree. To compensate for the errors observed near the extremes of the range of motion, a correction method was evaluated that builds on a continuous piecewise model. This correction resulted in a limited improvement in accuracy of 0.3 degree, which likely holds limited clinical relevance.

This study has several noteworthy limitations. First, the dependency on the accuracy of the single goniometer measurement performed during the onboarding process is a limitation of our approach. While the accuracy of the goniometer remains suboptimal, we have demonstrated that such single measurements in the mid-flexion range show good repeatability. In addition, this potential error remains constant through the patient’s recovery pathway, thus providing relative validity to day-to-day within-patient comparisons to track a patient’s recovery. The current algorithm is also limited by relying on a hinge model for the knee while assuming that the sensors are positioned in the sagittal plane by instructing the patients to apply the sensors on the lateral aspect of the thigh and shank respectively. In the literature, the flexion axis is often determined experimentally during a calibration maneuver, hence correcting for potential sensor misplacement errors [3,28]. The presented dynamic accuracy through the range of motion is however in line with results published in the literature, notwithstanding these published algorithms have often been validated with an optimized patient population (young adults). This suggests limited room for improvement by considering such corrections and provides confidence in the presented approach. Finally, this study is limited by the relatively small patient population and small number of involved clinicians. While the population used in this study was intended to be representative of the eventual end users for the IMU system, the inclusion of only eight subjects may not capture certain trends in sub-populations or edge cases where the accuracy may be lower. Additionally, the use of a few well-trained clinicians to perform the test likely yields best case results. When implemented in clinical practice, the accuracy of the methods presented here would be subject to human error, which may be more likely in less well-trained practitioners.

## 5. Conclusions

In conclusion, this paper has evaluated the accuracy of a skin-worn, wearable sensor system in measuring knee flexion. The system consists of two IMU sensors attached to the lower limb above and below the knee using a dual-patch system. During onboarding, the patient rests their operated leg on a box with a fixed height while the corresponding knee flexion angle is measured by a trained physiotherapist. Subsequently, the patient repeats this position after every sensor re-application (at home). This process has been shown to be repeatable, with a minimal mean absolute error of 1.6 degrees. The additional sensor error obtained as the patient moves through the range of motion is limited to a mean absolute error of 2.6 degrees. The combined error of 4.2 degrees allows for a clinically meaningful interpretation of the measurements, facilitating effective remote patient monitoring and potential tele-rehabilitation on a larger cohort of patients.

## Figures and Tables

**Figure 1 jfmk-06-00060-f001:**
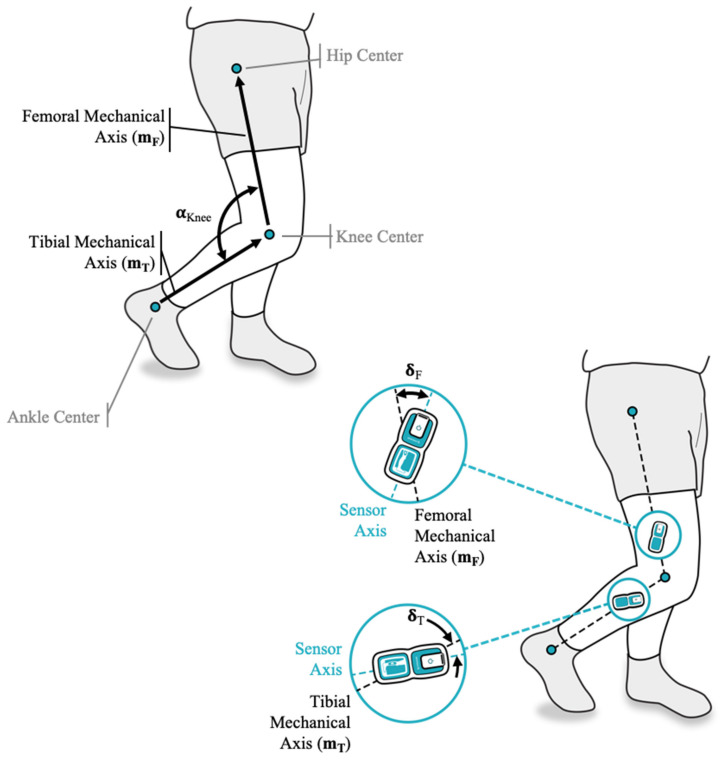
Position of thigh and shank sensor relative to the respective mechanical axes after sensor (re-)application is unknown resulting in varying offset angles δ_F_ and δ_T_.

**Figure 2 jfmk-06-00060-f002:**
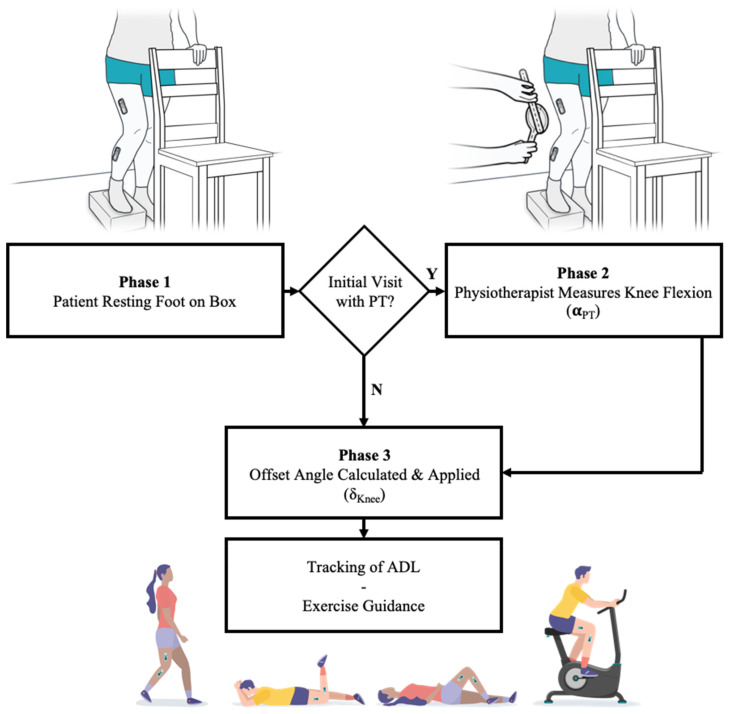
Establishing a sensor offset angle to facilitate accurate knee flexion measurements by having the patient stand on a box. Initially a physiotherapist measures the associated knee flexion angle. This angle while standing on a box is used to determine a daily sensor offset angle which corrects the sensor readings during activities of daily living or guided exercises.

**Figure 3 jfmk-06-00060-f003:**
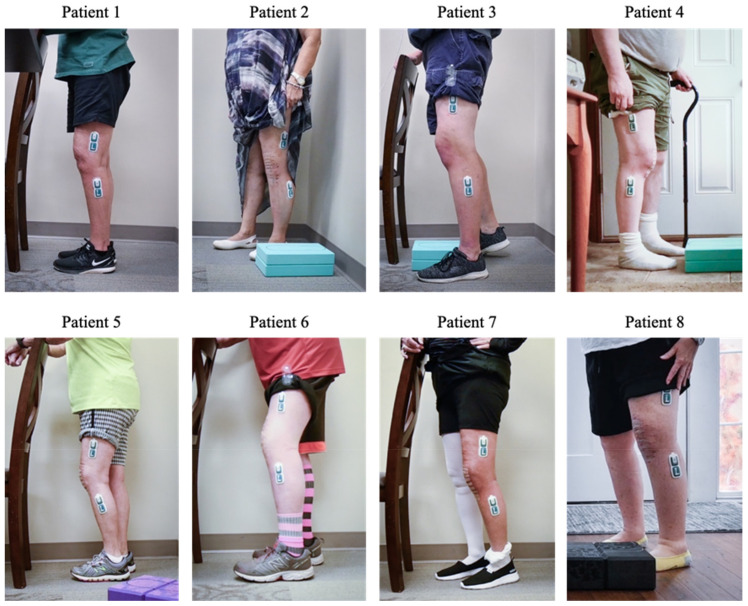
Summary view of sensors mounted on participants’ legs during the onboarding visit.

**Figure 4 jfmk-06-00060-f004:**
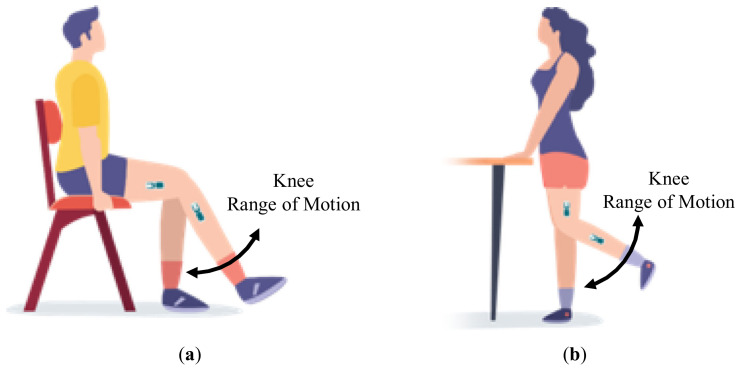
Schematic representation of: (**a**) long arc quad exercises and (**b**) standing knee bends.

**Figure 5 jfmk-06-00060-f005:**
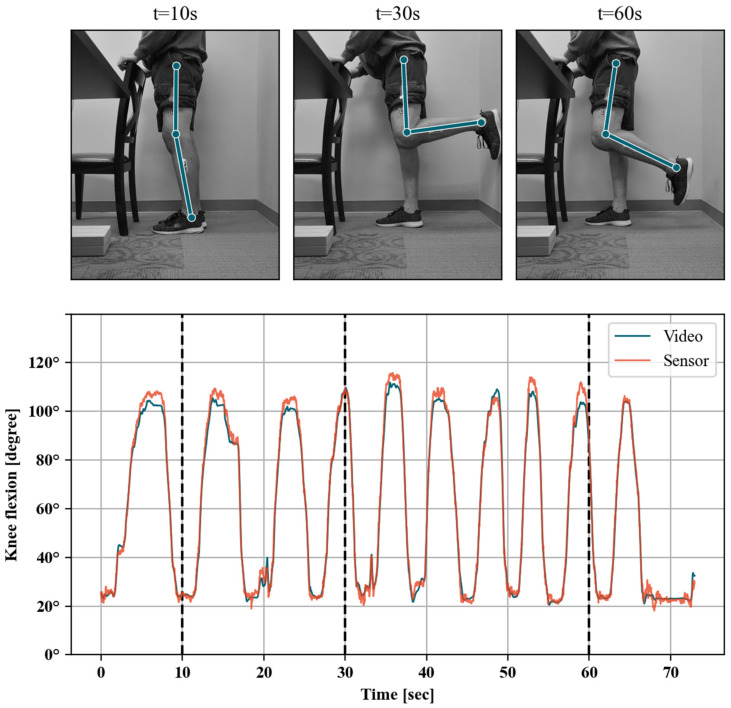
Example time-aligned timeseries of sensor and video knee angle flexion measurements (**bottom panel**). Joint locations estimated by the video system are shown for three example video frames (**top panel**).

**Figure 6 jfmk-06-00060-f006:**
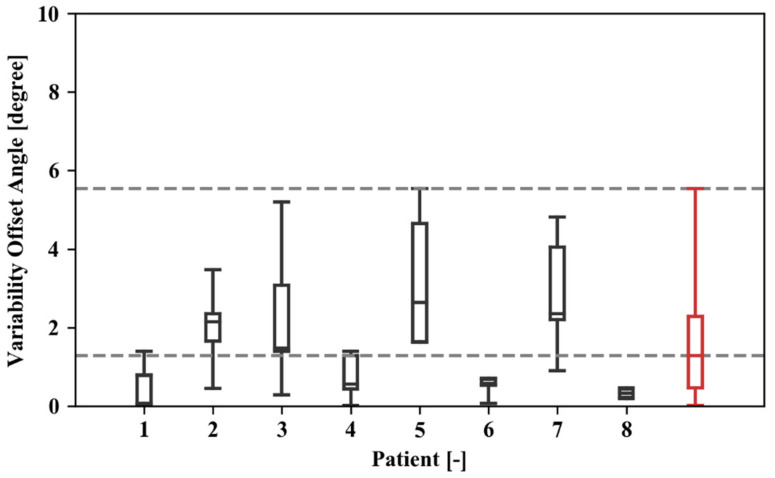
Box and whisker plot showing the distribution of the absolute offset angles over various repeats for different study participants and the overall results for all test data (in red). The upper and lower limit for each of the boxes represent the 25th and 75th percentiles of the observations for a given dataset, whereas the extreme values are bound by the whiskers.

**Figure 7 jfmk-06-00060-f007:**
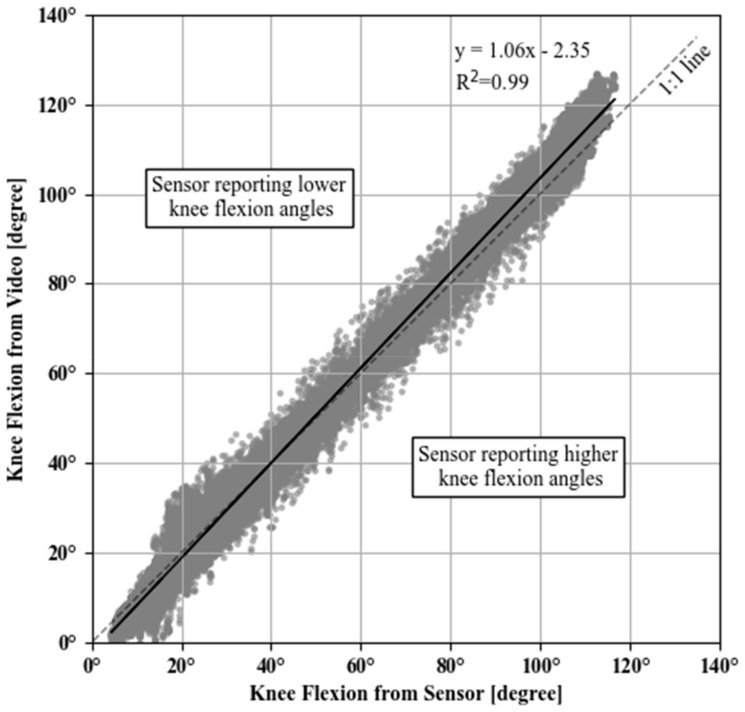
Scatter plot between sensor- and video-measured knee flexion angles for all sessions. Black line corresponds to the fitted linear regression with sensor measurements as an independent variable (x) and video angle measurements as the dependent (y) variable. The diagonal dashed line (1:1 line) indicates the equality of the two measurements.

**Figure 8 jfmk-06-00060-f008:**
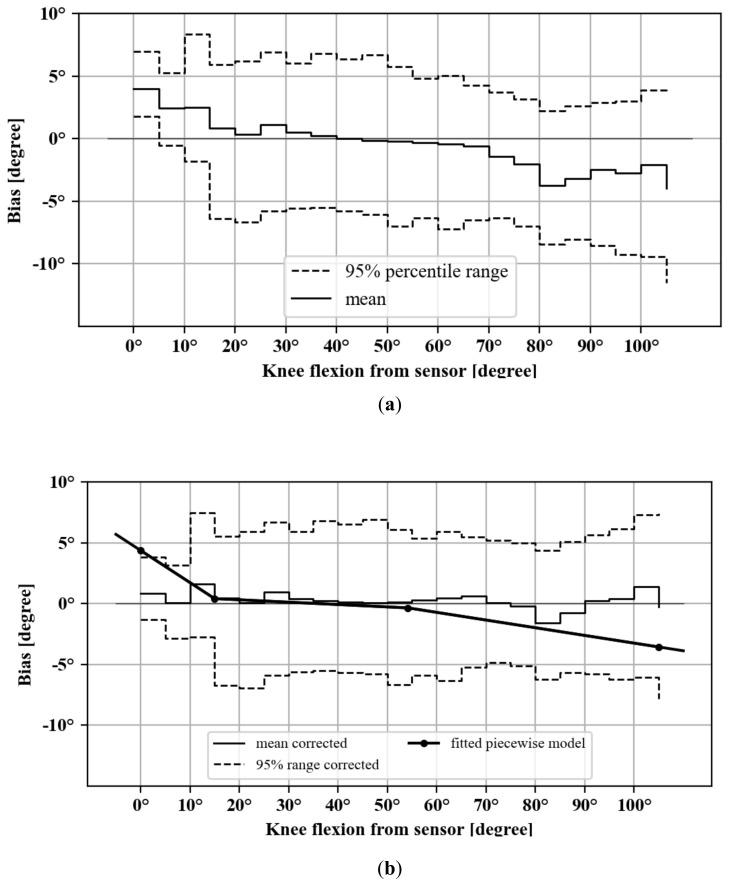
Sensor knee flexion angle bias in 5-degree bins, with 95-percentile range for: (**a**) uncorrected and (**b**) corrected measurements. Also included is the fitted linear piecewise model with breakpoints (line with dotted markers in part **b**).

**Table 1 jfmk-06-00060-t001:** General demographic data for participating total knee patients.

#	Gender	Age (year)	Weight (kg)	Height (cm)	BMI (kg/m^2^)	Days Since Surgery at Onboarding	Video Analysis
1	Female	66	65	165	23.9	42	N
2	Female	61	93	163	35.0	21	N
3	Male	58	116	193	31.1	42	Y
4	Male	56	104	175	34.0	23	Y
5	Female	67	64	155	26.6	21	Y
6	Male	68	102	182	30.8	21	Y
7	Female	62	72	163	27.1	21	Y
8	Female	67	70	159	27.7	26	N

**Table 2 jfmk-06-00060-t002:** Difference between goniometer measurements at subsequent clinic visits for the participating patients.

Patient #	1	2	3	4	5	6	7	8
**Goniometer Angle Initial Visit [degree]**	60	65	60	58	59	60	74	60
**Goniometer Angle Final Visit [degree]**	62	63	62	50	56	56	74	60

**Table 3 jfmk-06-00060-t003:** Repeatability of offset angle while patient is taking a pose on the box.

Patient #	1	2	3	4	5	6	7	8	All
Number of Repeats (n)	5	5	5	5	5	5	5	4	39
Mean Absolute Variability (degree)	0.6	2.0	2.3	0.7	3.2	0.5	2.9	0.3	1.6
Max Variability (degree)	1.4	3.5	5.2	1.4	5.5	0.7	4.8	0.5	5.5

**Table 4 jfmk-06-00060-t004:** Bias statistics for each data logging/exercise session. Duration is the length of time aligned between video and sensor measurements.

Patient	Visit 1: First 2: Final	Exercise	Duration [sec]	Bias [degree]		
				Mean	Mae	Std
**3**	2	Long Arc Quads	57.17	1.80	1.26	1.91
	2	Long Arc Quads	42.48	0.41	1.14	1.91
	2	Standing Knee Bends	133.37	0.16	2.57	3.01
	2	Standing Knee Bends	72.96	−0.99	1.93	2.40
	2	Standing Knee Bends	96.13	1.58	2.40	2.87
	2	Standing Knee Bends	92.78	−1.43	1.68	2.29
**4**	2	Long Arc Quads	117.15	1.56	1.86	2.75
	2	Standing Knee Bends	80.73	−3.55	3.46	4.15
	2	Standing Knee Bends	64.36	−3.50	2.96	3.62
	2	Standing Knee Bends	87.10	−3.04	3.98	4.65
	2	Standing Knee Bends	106.99	−4.73	2.96	3.70
	1	Standing Knee Bends	64.18	−1.98	3.22	3.85
**5**	2	Long Arc Quads	108.17	1.49	1.39	1.87
	2	Standing Knee Bends	111.28	−0.97	3.40	3.87
	1	Standing Knee Bends	29.25	−0.87	2.40	3.22
	1	Standing Knee Bends	132.82	−0.79	1.82	2.27
**6**	2	Standing Knee Bends	150.40	−0.48	4.41	4.91
	2	Standing Knee Bends	174.92	−0.37	3.33	4.07
	2	Standing Knee Bends	185.19	−0.05	2.29	2.84
**7**	1	Standing Knee Bends	157.39	0.75	2.35	2.79

**Table 5 jfmk-06-00060-t005:** Bias statistics in 5-degree bins, including after applying the modeled linear piecewise correction.

Knee Flexion (degree)	Uncorrected Bias			Linear Piecewise Corrected Bias		
	Mean	Mae	Std	Mean	Mae	Std
**0–5**	3.98	4.95	1.61	0.74	1.29	1.61
**5–10**	2.45	1.78	1.45	0.12	1.27	1.57
**10–15**	2.51	2.80	2.65	1.90	2.40	2.64
**15–20**	0.84	2.17	3.01	0.67	2.28	3.03
**20–25**	0.34	2.43	3.23	−0.06	2.51	3.22
**25–30**	1.12	2.91	3.30	0.54	2.61	3.31
**30–35**	0.48	2.61	2.87	0.03	2.21	2.87
**35–40**	0.21	2.58	2.97	0.02	2.23	2.97
**40–45**	0.00	2.55	2.99	0.07	2.24	3.00
**45–50**	−0.16	2.58	3.08	0.17	2.24	3.08
**50–55**	−0.21	2.48	3.09	0.39	2.29	3.09
**55–60**	−0.31	2.20	2.71	0.54	2.00	2.71
**60–65**	−0.42	2.13	2.97	0.68	2.21	2.97
**65–70**	−0.61	1.90	2.68	0.76	2.03	2.68
**70–75**	−1.44	2.11	2.55	0.18	1.92	2.55
**75–80**	−2.03	2.52	2.76	−0.14	2.15	2.76
**80–85**	−3.75	3.74	2.74	−1.58	2.64	2.72
**85–90**	−3.20	3.14	2.72	−0.83	2.35	2.74
**90–95**	−2.49	2.49	2.99	0.14	2.29	2.97
**95–100**	−2.73	2.61	3.39	0.19	2.75	3.41
**100–105**	−2.07	2.28	3.79	1.12	3.25	3.81
**105–110**	−3.95	3.12	4.10	−0.55	3.29	4.08
**Overall Mean**	**−0.52**	**2.64**		**0.23**	**2.29**	

**Table 6 jfmk-06-00060-t006:** Parameters of fitted piecewise regression model.

Sensor Knee Angle Range	Intercept	Slope
0°–15°	4.43	−0.23
15°–47°	0.51	−0.01
47°–105°	2.71	−0.06

## Data Availability

Restrications apply to the availability of the data. Data was obtained from Oklahoma Joint Reconstruction Institute and EnMovi Ltd. and are available from the authors with permission from Oklahoma Joint Reconstruction Institute and EnMovi Ltd.

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
