# Peer review of "Accuracy of Measuring Knee Flexion after TKA through Wearable IMU Sensors"

_jfmk, 2021, doi:10.3390/jfmk6030060_

Round 1
Reviewer 1 Report
In the presented manuscript, the authors investigate the accuracy of wearable IMU-sensors in patients after TKA. The authors correlate sensor measurements at various time points with video and physiotherapists measurements to demonstrate usefulness in the investigated sensors for at-home monitoring of knee range of motion after surgery.
While the presented data seems to show the high accuracy of IMU-sensors, the manuscript falls short on providing data showing the usefulness and accuracy for at-home measurements done by patients. Additionally, neither the clinical relevance is sufficiently explained nor is the sample size large enough to draw definite answers from.
Major points
- Introduction: Clinical relevance of follow-ups is not clearly stated. The source cited [2] lists pain and the amount of physiotherapy as major patient outcome variables; neither of which is investigated in the current study
- The authors conduct their study on only 8 patients with largely varying height and BMI. The study population seems much too small to draw any conclusions from and should be significantly increased
- It is unclear why only 5 patients have been examined using the gold (video) standard. The authors should repeat gold standard measurements for all patients involved in this study.
- Can the authors elaborate on the number of repeats in patients with high max variability (Table 3)? Why did the authors repeat measurements in these patients more often than in others? It seems like the goal was to lower mean the absolute variability in these patients and not accurately display the true data. All patients need the same number of repeats (n) for better evaluation of accuracy.
- The authors mention at-home measurements several times throughout the manuscript and claim to investigate usability and accuracy in this setting. However, the manuscript fails to provide any such data. Please provide these continuous at home measurements.
- Thus, it is also unknown if the sensor attachment by patients at home was equally sufficient compared to in a clinical setting.
- In the introduction, the authors mention overweight in patients and sensor attachment by patients as one of the main challenges. Neither was investigated in this manuscript but is highly relevant both for the sensors patients reattach and those that stay on the patient throughout the investigation. Authors should provide data on these two issues.
- The authors should add a paragraph discussing the clinical relevance of daily measurements. How does this granularity improve outcome?
Minor points
- Figure numbering is not correct in some places
- Is RMSE not more appropriate than MAE in the study analysis?
Author Response
see attached document

Reviewer 2 Report
The main problem with the manuscript is scientific writing, which is also too long to read.
Introduction
In my opinion, the description of the technical details and apparent facts should be shortened considerably. 
Three paragraphs would be appropriate (the importance of measuring the joint angle, modalities of measuring and its included problem, and study purpose/hypothesis).
LN58
What does “the gold standard” mean? It should be defined more clearly.
The authors should state their aim of study and hypothesis clearly.
In LN88-92, they described the study purpose. But, what “how to repeatably establish the…” mean?
Ln 92
RoM should be read ROM.
Methods
The technical terms are hard to understand and need a definition (e.g. mt resp).
I could not understand the position of the sensor during the sessions. It should be described.
In table 1, Pts 1, 2, and 8 did not complete the video analysis. How did the authors validate the data?
Results.
LN269
What are the linear models mean? There was no description of the statistics in the method section.
What is R2 mean? Coefficient of determination? I could not believe there is an R2 > 0.99 in a linear regression model.
Discussion
The first paragraph should be the answer to the study hypothesis.
The majority part of the introduction should move to the discussion section.
References should be described as instructions for authors.
Reviewer 3 Report
I think the work is well written and presented, and it shows a real novelty.
Conclusions are plausible, although the sample is limited in number and probably should be more varied (age range and physical activity).
I believe it is useful, if possible, to report the physioterapic scheme operated to determine possible errors, underlining whether it was the same for all subjects or not.
The error bar in fig 6 is large, do you think it is possible to improve it? Also in future works?
Please check tab 4, second column, all data are correct?
Round 2
Reviewer 1 Report
The authors addressed most of my concerns.
I would advise the authors to change the table and figure in the manuscript according to the authors response no 4.
Also please check the figure numbering (Figure 5 twice in the manuscript)
Author Response
Thanks for your comments, this has been updated accordingly.
Reviewer 2 Report
The manuscript has been revised well. I think this manuscript will be acceptable.
Author Response
Thank you for your time and feedback in reviewing our manuscript.